# Risk Factors Analysis for 90-Day Mortality of Adult Patients with Mild Traumatic Brain Injury in an Italian Emergency Department

**DOI:** 10.3390/geriatrics9020023

**Published:** 2024-02-23

**Authors:** Daniele Orso, Giulia Furlanis, Alice Romanelli, Federica Gheller, Marzia Tecchiolli, Franco Cominotto

**Affiliations:** 1Department of Anesthesia and Intensive Care, ASUFC University Hospital of Udine, Via Pozzuolo 330, 33100 Udine, Italy; 2Department of Emergency Medicine, ASUGI University Hospital of Trieste, 34100 Trieste, Italy; giulia.furlanis@asugi.sanita.fvg.it (G.F.); franco.cominotto@asugi.sanita.fvg.it (F.C.)

**Keywords:** head trauma, prediction, traumatic brain injury, emergency department, mortality

## Abstract

**Purpose**: The most prominent risk factors for mortality after mild traumatic brain injury (TBI) have not been established. This study aimed to establish risk factors related to 90-day mortality after a traumatic event. **Methods**: A retrospective cohort study on adult patients entering the Emergency Department of the University Hospital of Trieste for mild TBI from 1 January 2020 to 31 December 2020 was conducted. **Results**: The final population was 1221 patients (median age of 78 years). The 90-day mortality rate was 7% (90 patients). In the Cox regression model (likelihood ratio 110.9; *p* < 2 × 10^−16^), the variables that significantly correlated to 90-day mortality were age (less than 75 years old is a protective factor, HR 0.29 [95%CI 0.16–0.54]; *p* < 0.001); chronic liver disease (HR 4.59 [95%CI 2.56–8.24], *p* < 0.001); cognitive impairment (HR 2.76 [95%CI 1.78–4.27], *p* < 0.001); intracerebral haemorrhage (HR 15.38 [95%CI 6.13–38.63], *p* < 0.001); and hospitalization (HR 2.56 [95%CI 1.67–3.92], *p* < 0.001). Cardiovascular disease (47% vs. 11%; *p* < 0.001) and cognitive impairment (36% vs. 10%; *p* < 0.001) were more prevalent in patients over 75 years of age than the rest of the population. **Conclusions**: In our cohort of patients with mild TBI, 90-day mortality was low but not negligible. The risk factors associated with 90-day mortality included age, history of chronic liver disease, and cognitive impairment, as well as evidence of intracerebral hemorrhage and hospitalization. The mortality of the sub-population of older patients was likely to be linked to cardiovascular comorbidities and neurodegenerative diseases.

## 1. Introduction

Traumatic brain injury (TBI) is characterized by an external force that causes an alteration in brain function or other evidence of brain pathology [1]. TBI is estimated to cause an estimated 10 million deaths or hospitalizations worldwide each year [2]. More than 1.7 million cases of TBI occur annually in the United States. About 75% of this group are classified as having mild TBI [3].

TBI has traditionally been classified using the Glasgow Coma Scale (GCS), a clinical score that measures consciousness level. In general, mild TBI is diagnosed in patients with a GCS score of 13–15; moderate TBI is diagnosed in those with a GCS score of 9–12; and severe TBI is diagnosed in those with a GCS score < 8 [2].

The geriatric population (adults ≥ 75 years) is particularly susceptible to TBI due to several concomitant risk factors: alteration of balance and reduced postural reflexes, musculoskeletal limitations, reduction in visual acuity, and intake of drugs that can alter alertness or cause orthostatic hypotension [4]. Despite the development of guidelines proposed by some emergency medicine scientific societies [5,6], clinical research has not yet established the most prominent risk factors associated with mortality after mild TBI. This determination would have a considerable clinical implication: the need to subject many patients to a computerized tomography (CT) scan in the face of a rare finding of brain lesions, constituting a conspicuous workload for all the emergency departments (EDs) in the Western world (where the elderly population is increasing). The CT scan, especially when repeated for the serial evaluation of victims of head trauma, determines the exposure to ionizing radiation. It also establishes a commitment by the Radiology Department to meet urgent requests from an ever–growing population of patients in the ED [7]. Besides these factors, we need to determine the risk factors that lead to an increased mortality rate after a mild TBI. This finding could have implications for the most effective clinical measures to manage this patient cohort.

The aim of this study was to establish risk factors related to 90-day mortality after a traumatic event.

## 2. Materials and Methods

### 2.1. Study Design

This study was a retrospective cohort study utilizing an internal database of all head trauma cases in the ED of the University Hospital of Trieste from 1 January 2020 to 31 December 2020. The University of Trieste Institutional Review Board exempted this study from review. The study adhered to the international and national regulations established by the Declaration of Helsinki. Each patient was asked for consent to process personal data and use it for care purposes.

### 2.2. Study Setting and Population

Patients were enrolled consecutively upon admission to the ED of the University Hospital of Trieste (a census of about 75,000 patients per year) in Northeast Italy. The staff dedicated to inserting clinical data into the database had neither management nor decision-making roles in handling the enrolled patients.

The inclusion criteria were adult patients (≥18 years old) entering the ED for minor head trauma (GCS score ≥ 13) who underwent a brain CT scan. Pregnant patients were excluded.

### 2.3. Study Variables

The recorded variables were age; sex; date and time of ED visit; the time of the first head CT scan; first head CT scan report; comorbidities (hypertension, diabetes mellitus, chronic obstructive pulmonary disease [COPD], chronic alcoholism, chronic liver disease or kidney disease, cognitive impairment, and history of relapse falls in the previous 12 months); use of antiplatelet and anticoagulant drugs (vitamin K antagonists [VKAs], direct oral anticoagulants [DOACs], and heparin or analogues in anticoagulant dosage [LWMH]—fondaparinux, aspirin, and clopidogrel); INR and platelet count; clinical evaluation at the onset (GCS, amnesia, loss of consciousness, vomiting, headache, focal neurological deficits, and confusion); high energy impact (hit pedestrian, fall from >2 m, and ejected passenger); the presence of skeletal fractures; and patient management (hospitalization, discharge, self-discharge, or abandonment). Patients were followed up for 90 days after the TBI. The patient was observed to either be admitted to a hospital ward or discharged after an observation period in the emergency department, usually 6–12 h. The term “hospitalization” is reserved for patients who have been admitted to the hospital.

### 2.4. Age-Based Population Analysis

We divided the sample based on age and, in particular, categorized two sub-populations (those who are more or less than 75 years old) in order to analyze the differences between the so-called ‘young’ and ‘old’ populations. The two sub-populations were compared to determine the statistical significance of the variables.

### 2.5. Statistical Analysis

The variables were expressed as median values (interquartile range; IQR) or frequency (%). Univariate hazard ratios (HRs) were computed from a Cox model where the grouping variable is introduced as an integer variable predictor. The *p*-values for hazard ratios are computed using the log-rank or Wald test under a Cox proportional hazard regression when the row variable is categorical or continuous, respectively. An alpha error of ≤0.05 (*p*-value) was considered statistically significant.

A multivariable Cox proportional hazards survival analysis was conducted to detect variables significantly related to 90-day mortality. To verify the assumption of the Cox regression inherent in hazards being time-invariant, we verified that the coefficient of the corresponding set of scaled Schoenfeld residuals with time was zero. In addition, the linearity assumption was confirmed by inspection of the Martingale residuals plotted against the continuous covariates. In case the hazards proved not time-invariant, we estimated the slope of the covariates over time by testing Aalen’s additive regression.

A Kaplan-Meier survival analysis was conducted to compare survival times among the significantly correlated variables with 90-day mortality. A log-rank test was run to determine if there were differences in the survival distribution for the different subgroups. A Bonferroni correction was made with statistical significance accepted at the *p* < 0.017 level in the multivariable model.

The Shapiro-Wilk test was used to verify the normal distribution in the comparison analysis between the two sub-populations. Kruskall-Wallis tests for continuous variables were used to compare non-normal distributions. Categorical variables were compared using the chi-squared or exact Fisher test when the expected frequencies were less than 5 in some cells. The Benjamini-Hochberg method was used to correct the multiplicity of *p*-value for pairwise comparisons between more than two categories. For every row variable, a risk ratio (RR) was determined.

Statistical analysis was performed using the R environment (version 4.1.2, R Foundation for Statistical Computing. Vienna, Austria).

## 3. Results

During the period considered, 1233 patients were enrolled. Of these, 12 were excluded because of incomplete reported data or the lack of informed consent. The final population was 1221 patients (Figure 1). The 90-day mortality rate was 7% (90 patients). Patients with mild TBI who died within 90 days had a median survival time of 45 days.

The median age was 78 years (interquartile range: 58–85). A total of 542 patients were male (44%). The most common comorbidities were hypertension (553; 45%), chronic heart disease (370; 30%), and cognitive impairment (301; 25%). In total, 191 patients (16%) took anticoagulation drugs and 248 patients (20%) took antiplatelet drugs. Additionally, 116 patients had a pathological finding (neurocranial fracture; subdural hematoma; epidural hematoma; subarachnoid hemorrhage; intracerebral hemorrhage; or hemorrhagic petechiae) on the head CT scan (10%). In 16 cases, the brain lesion was found on a second head CT scan. No patients with documented brain lesions underwent neurosurgery. The most frequent brain lesions on the head CT scan were a subdural hematoma (30; 2%), followed by a subarachnoid hemorrhage (29; 2%). 

The distribution of clinical variables with their evaluation as mortality risk factors are shown in Table 1.

In univariate survival analysis (Table 1), the statistically significant variables for 90-day mortality were age (median age of 88 for deceased patients and 77 for surviving patients; HR 1.07 [1.05–1.09]); a positive history of hypertension (HR 1.85 [95%CI 1.21–2.82]); diabetes mellitus (HR 2.17 [95%CI 1.37–3.44]); chronic kidney disease (HR 2.38 [95%CI 1.43–3.95]); chronic obstructive pulmonary disease (HR 2.26 [95%CI 1.20–4.25]); chronic liver disease (HR 4.12 [95%CI 2.37–7.18]); frequent falls (HR 2.59 [95%CI 1.71–3.94]); and cognitive impairment (HR 3.83 [95%CI 2.53–5.81]). Other variables related to an increased risk of death at 90 days were low-molecular-weight heparin (HR 3.81 [95%CI 1.20–12.0]); confusion following head trauma (HR 2.98 [95%CI 1.85–4.79]); and proximal fractures of lower limbs (HR 2.52 [95%CI 1.16–5.44]). There was no statistically significant correlation between mortality at 90 days and the amount of comorbidity (*p*-value 0.177) or indiscriminate anticoagulant or antiplatelet administration (*p*-value = 0.190).

As concerns the brain lesions found at the CT scan, only the cerebral hemorrhage (HR 11.5 [95%CI 4.68–28.4]) was significantly correlated with mortality at 90 days. Hospitalization was associated with an increased mortality at 90 days (HR 3.17 [95%CI 2.09–4.81]) (Figure 2, Figure 3 and Figure 4).

The Cox regression model (concordance 0.8 [se 0.02]; likelihood ratio 110.9; Wald test 113; score (log-rank) 156.1; *p* < 2 × 10^−16^) shows that variables significantly correlated to 90-day mortality were age (less than 75 years old is a protective factor, HR 0.29 [95%CI 0.16–0.54]; *p* < 0.001); chronic liver disease (HR 4.59 [95%CI 2.56–8.24], *p* < 0.001); cognitive impairment (HR 2.76 [95%CI 1.78–4.27], *p* < 0.001); intracerebral hemorrhage (HR 15.38 [95%CI 6.13–38.63], *p* < 0.001); and hospitalization (HR 2.56 [95%CI 1.67–3.92], *p* < 0.001) (Figure 5).

However, the presence of intracerebral hemorrhage (Chi^2^ 4.81, *p* = 0.028), hospitalization (Chi^2^ 6.16; *p* = 0.013), and cognitive impairment (Chi^2^ 4.90; *p* = 0.002) varies over time (Ljung-Box global test for the Cox model: Chi^2^ 15.2, *p* = 0.001). In the Aalen test (Chi^2^ 60.2; *p* = 1.1 × 10^−11^), each of the model variables exhibits a progressive slope over time. In particular, hospitalization, chronic liver disease, and cognitive impairment show an exponential increase in mortality over time (Figure 6).

### Age-Based Population Analysis

Analyzing the two sub-populations established through cut-off by age corresponding to 75 years, we found that the elderly population had a greater number of female individuals (66% vs. 43%; *p* < 0.001), a greater rate of comorbidity (hypertension: 66% vs. 21%, *p* < 0.001; diabetes mellitus: 21% vs. 9%, *p* < 0.001; chronic kidney disease: 18% vs. 4%; *p* < 0.001; chronic obstructive pulmonary disease: 8% vs. 4%; *p* = 0.005; cardiovascular disease: 47% vs. 11%; *p* < 0.001; cognitive impairment 36% vs. 10%; *p* < 0.001; and more history of falls: 30% vs. 13%; *p* < 0.001); a higher proportion of patients taking anticoagulants and/or antiplatelet drugs (50% vs. 16%; *p* < 0.001); a higher INR median value (1.12 [95%CI: 1.03–1.56] vs. 1.03 [95%CI 0.97–1.09]; *p* < 0.001); more frequently proximal fractures of the limbs (5% vs. 2%; *p* = 0.026 for the superior limbs; 5% vs. 2%; *p* = 0.002 for the inferior limbs); and confusion (14% vs. 7%, *p* < 0.001) than the younger population. On the other hand, the younger population was more likely to have a history of chronic alcoholism (15% vs. 1%; *p* < 0.001). They also more often experienced headaches (13% vs. 2%, *p* < 0.001) and loss of consciousness (11% vs. 5%, *p* < 0.001). Finally, they more often presented a high-energy trauma (9% vs. 2%, *p* < 0.001).

Analyzing the older population, it is noted that patients who died at 90 days, were significantly older (87 [95%CI 82–91] vs. 84 [95%CI 81–89]; *p* < 0.001; RR 1.04) and they had a greater number of comorbidities (3 [95%CI 2–4] vs. 2 [95%CI 1–3]; *p* < 0.001), in particular a greater rate of cognitive deterioration (55% vs. 34%; *p* < 0.001; RR 2.17), chronic liver disease (12% vs. 3%; *p* = 0.002; RR 3.24), and history of falls (43% vs. 28%; *p* = 0.011; RR 1.79). In addition, they suffered from confusion more often (22% vs. 12%; *p* = 0.002; RR 1.72) at the first medical examination.

However, there were some differences in variables when comparing only those who died to the two sub-populations of younger and older (74 vs. 13 deaths, respectively; the age was not provided in three cases). The incidence of chronic liver disease (39% vs. 12%; *p* = 0.032; RR 3.26) and chronic alcoholism (39% vs. 0%; *p* < 0.001; RR 10.2) were more prevalent in the younger population. The elderly were more susceptible to hypertension (70% vs. 15%; *p* < 0.001; RR 0.11) and cardiovascular disease (45% vs. 0%; *p* = 0.001; RR 0) (Figure 7).

## 4. Discussion

In our cohort of patients with mild TBI, 90-day mortality was 7%. Our finding is in line with the recent literature [8,9,10]. Grosswasser et al., in a small cohort (279 patients) of TBI victims followed for a 5-year period, reported a mortality rate of 11.5% [9]. Leitner et al. reported a 30-day mortality rate of 2.5% in a population similar to ours (1788 patients), but almost 6% in the population over the age of 65 [10]. Moreover, Ottochian et al. found a mortality of 6% [11], and McMillian et al. found a mortality of about 5% per year [12]. Popp et al. found a greater mortality (about 38%) in elderly patients with post-traumatic brain lesions [13]. However, the authors also enrolled patients with severe TBI. Castillo-Angels et al. detected a mortality rate of approximately 2%. Also, they did not find a difference between patients with and without brain lesions [14]. Concerning brain lesions associated with 90-day mortality, we found that only intraparenchymal cerebral hemorrhage was a risk factor. In contrast, other lesions (more frequently found), such as subdural hematoma or subarachnoid bleeding, were not associated with an increased risk of death at 90 days.

It is known from the literature that patients with a documented brain lesion are hospitalized more frequently than patients with a negative head CT scan [10]. The rate of hospitalization in our population was 20%. Although direct damage from head trauma may have been the cause of hospitalization, none of the hospitalized patients underwent neurosurgery. While there were 30 cases of subdural hematoma found, only 3 of them died within 90 days, even though none of them had undergone neurosurgery. Small case studies in the literature show that patients with subdural hematoma from ground-level falling have a higher mortality rate than those with high-energy trauma cases [15]. The connection between non-operational issues caused by the fragility of the elderly population and ageism is uncertain. The selection of patients for surgery is a controversial aspect that has a significant clinical impact.

Furthermore, although the literature does not show robust evidence of the need to repeat the CT exam after a first negative CT scan, patients with mild TBI are likely to be stationed in the ED for a second head CT scan [16,17]. In the literature, a worsening neurological status appears to be the major predictor for a second head CT scan with pathological findings [18,19]. In this respect, our data suggest that the careful stratification of the risk of complications based on medical history, imaging findings, and clinical performance may help reduce hospitalization and save resources that may not improve outcomes [20,21].

In fact, we found that, in addition to age and clinical history, hospitalization is a 90-day mortality risk factor. Given our study design, we cannot establish how much these data are attributable to a causal link (e.g., for infectious complications) or are due to the patient’s fragility. It is noteworthy that our population could be particularly fragile, as indirectly suggested by the high percentage of patients with cognitive impairment. In addition, studies have shown that TBI can worsen cognitive impairment in the medium-to-long-term [22,23,24]. Pentland et al. found a delayed mortality rate resulting from head trauma due to alcohol abuse, suicidal attempts, or repeated trauma [25].

The absence of a significant correlation between the degree of comorbidity in the population is intriguing. However, this is consistent with the literature. Recently, a systematic review by Xiong et al., analyzing 27 cohort studies, found no significant correlation between the comorbidity condition load and outcome [26]. A more recent systematic review by Hanafy et al. is in agreement with this result [27]. As the authors point out, the high heterogeneity of the methods used to report the load of comorbidity conditions makes it difficult to draw conclusions. The standardization of the comorbidity degree report remains an open question in TBI studies. Unfortunately, for the study design that we have set, we are not able to better detail the degree of comorbidity of the sample we analyzed (e.g., Charlson comorbidity index, etc.). However, in our population, we observe that the comorbidities between the elderly and the young were qualitatively distinct. Alcohol-related disorders, including chronic liver disease, were more prevalent among the younger population, while cardiovascular disease was more prevalent among the older population. From a young age, alcohol consumption is linked to TBI, as evidenced in [28]. This is connected to a higher rate of high-energy accidents in younger patients. The elderly population, on the other hand, is more vulnerable to cardiovascular diseases and cognitive decay [29].

Otherwise, from analyzing the older population, it appears that the number of comorbidities was significantly correlated with 90-day mortality. In particular, cognitive impairment and a history of falls were factors involved in the outcome of this population. The fragility of these patients seemed to determine the outcome more than the trauma itself. According to some studies, mild TBI is a significant risk factor for death among the elderly population [30]. From a molecular viewpoint, TBI appears to be an inflammatory trigger that causes neurodegeneration [31]. It seems that the process of accelerated neuronal senescence is the cause of cognitive decline. In turn, this factor can lead to a vicious circle of falls and more traumatic events.

However, it is worth exploring the fact that age itself (even without the contribution of comorbidities) is a risk factor for mortality in the event of TBI. Ostermann et al. discovered that respiratory failure was a risk factor for short-term mortality in elderly (>65 years) TBI patients [32]. In view of the clinical fragility of this population, and in light of the role of hospitalization in determining the risk of death, it is plausible that conditions related to hospitalization (e.g., nosocomial infections) can contribute to 90-day mortality. Indeed, Selassie et al., in a retrospective study conducted on more than 41,000 patients, found that the occurrence of nosocomial infections (most of which are respiratory infections) are one of the causes of death of patients suffering from TBI [33]. However, it is a challenge to establish the epidemiology of elderly patients with mild TBI and comprehend the factors responsible for the unfortunate outcome, because the majority of the current literature focuses on severe or at least moderate TBI. The population who experiences mild TBI, particularly those who are elderly, seem to be receiving little attention from clinical studies.

We found no statistically significant differences in the outcome between females and males. Although the literature seems to describe a higher incidence of TBI in elderly women, there is no evidence that this could worsen the outcome of female patients [34]. However, it can be seen that the female gender is more represented in the older population. Aging seems to cause a selection towards the female gender; however, this presents significantly worse rates of trauma-specific survival [35].

We did not find any statistically significant difference in our population’s intake of anticoagulants or antiplatelet therapy. The most recent systematic reviews and meta-analyses found no statistically significant differences in the mortality rates of these subgroups of patients with mild TBI [36,37,38]. This result is significant because many patients with mild TBI, despite the absence of any clinical signs, are subjected to CT scans and an observation period exclusively due to chronic therapy with anticoagulants [39]. However, our findings show that anticoagulant and/or antiplatelet treatment did not affect the 90-day mortality rate. A recent retrospective study in Austria found that a 1-year increase in mortality can be attributed to both anticoagulants and antiplatelets [40]. However, this study also considered severe TBI. The low mortality rate found specifically in the cohort of patients with mild TBI makes it hard to draw conclusions. Furthermore, a large study by Fahkry et al., that analyzed the geriatric population (33,710 patients), found that anticoagulation and/or antiplatelet therapy did not have a significant correlation with mortality [41]. This may indicate that, in the determination of the outcome of this cohort of patients, other factors (such as those we have found) mostly predict an unfortunate outcome.

Although the literature shows a historically increasing trend in head CT scans for patients suffering from mild TBI [42,43], using more stringent clinical scores could probably achieve the most cost-effective strategy available to date [44,45,46]. As we can deduce from the results of our study, the adult population of victims of mild TBI is particularly uneven in terms of the risk of intracranial bleeding and the risk of complications and death [47]. Our findings show that some pre-existing conditions can contribute more to the outcome of these patients than the trauma injuries themselves. The elderly population is particularly susceptible to negative outcomes and therefore deserves specific management.

## 5. Limitations

In the predictive model, we included all the clinical variables detectable at the clinical examination in the Emergency Department. We cannot exclude that any collateral findings during the imaging process have a greater predictive role, albeit of little clinical utility for the emergency physician. According to the main international guidelines, the first head CT scan was performed within 6 h from the patient’s first evaluation. Any variation in clinical management could affect the outcome analyzed.

The cause of death for patients cannot be determined by us. The reason is partly because the cause of death for patients who are not admitted has not been explained and partly because even admitted patients often have generic and not sufficiently detailed causes of death. It came to our attention that 26 patients died within a week of the trauma. Despite the inability to establish a causal link, it is plausible that the trauma has played a role in determining the outcome.

Although adult patients aged 18 and over were included in the data collection, there were no representatives from 18 to 57. Younger people are likely to be more susceptible to moderate or severe TBI than older individuals.

## 6. Conclusions

In our cohort of patients with mild traumatic brain injury, mortality was 7%. Age, history of chronic liver disease, and cognitive impairment, along with evidence of intracerebral hemorrhage and hospitalization were found to be risk factors related to 90-day mortality. The mortality of the sub-population of older patients was likely to be linked to cardiovascular comorbidities and neurodegenerative diseases.

## Figures and Tables

**Figure 1 geriatrics-09-00023-f001:**
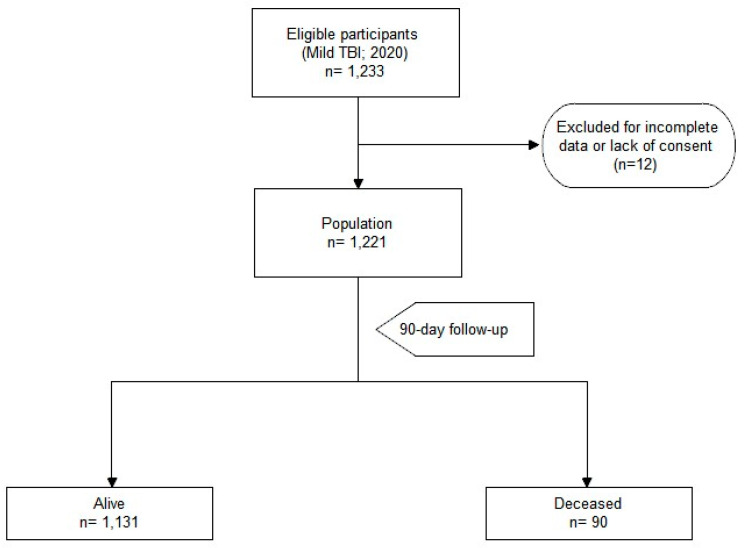
Enrollment process flow chart.

**Figure 2 geriatrics-09-00023-f002:**
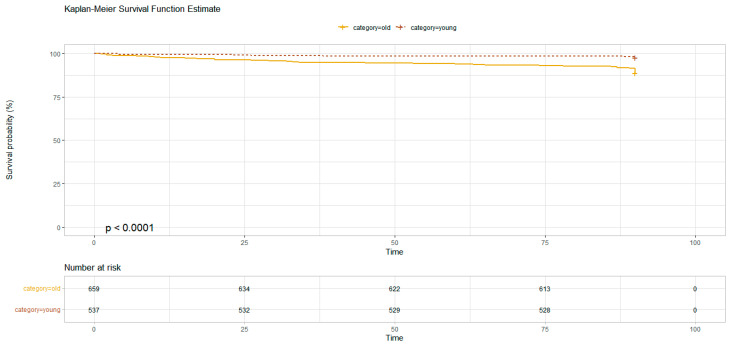
Kaplan-Meier survival function estimate plot for population stratified for ≥75 (solid line) and <75 (dotted line; Chi^2^ 33.7; *p* = 6 × 10^−9^).

**Figure 3 geriatrics-09-00023-f003:**
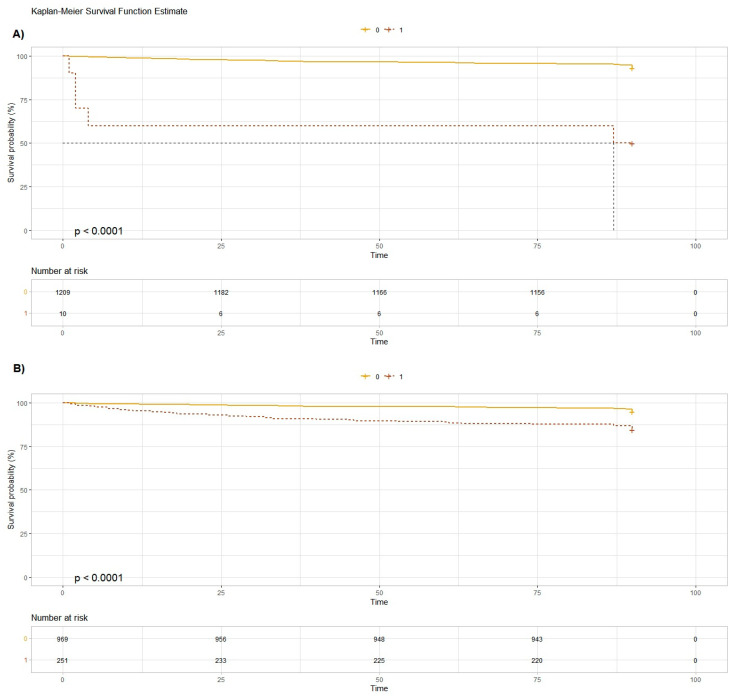
Kaplan-Meier survival function estimate plot for intracerebral hemorrhage (Plot **A**) and hospitalization (Plot **B**). The dotted line represents the patients with intracerebral hemorrhage and those that were hospitalized, respectively. Plot **A** indicates that survival rates decreased significantly in patients with intracerebral hemorrhage within the first few days (within the first 10 days). After 90 days of follow-up, the survival rate of these patients was 50% (Chi^2^ 45.1; *p* = 2 × 10^−11^). In Plot **B**, it was observed that the survival rate of hospitalized patients decreased over time as compared to non-hospitalized patients (Chi^2^ 32.9; *p* = 1 × 10^−8^).

**Figure 4 geriatrics-09-00023-f004:**
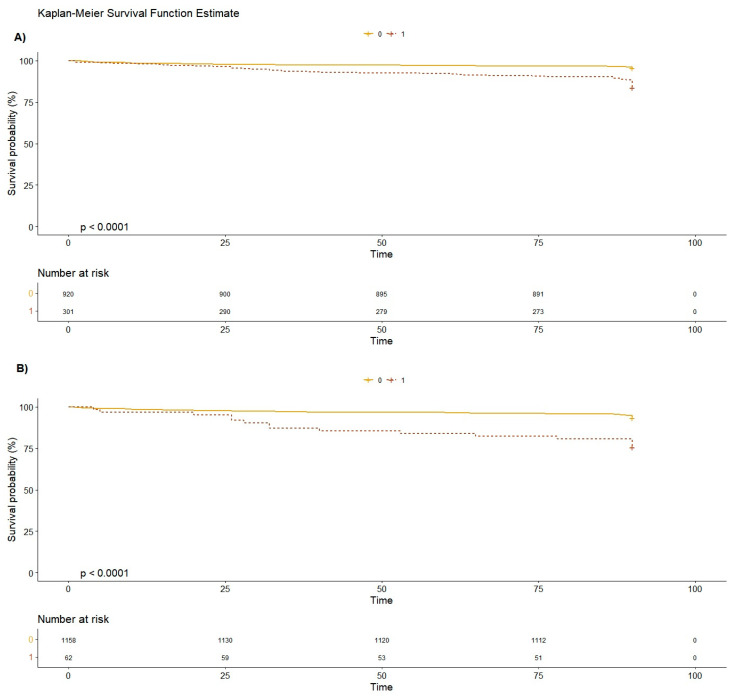
Kaplan-Meier survival function estimate plot for cognitive impairment (Plot **A**) and chronic liver disease (Plot **B**). The dotted line represents the patients with cognitive impairment and chronic liver disease, respectively. Plot **A** displays a slight but gradual decline in the survival rate to day 90 follow-up for patients with cognitive impairment, compared to those without cognitive impairment (Chi^2^ 46.5; *p* = 9 × 10^−12^). Plot **B** illustrates that patients with chronic liver disease experienced a decrease in survival rates from day 25 to day 90(Chi^2^ 29.4; *p* = 6 × 10^−8^).

**Figure 5 geriatrics-09-00023-f005:**
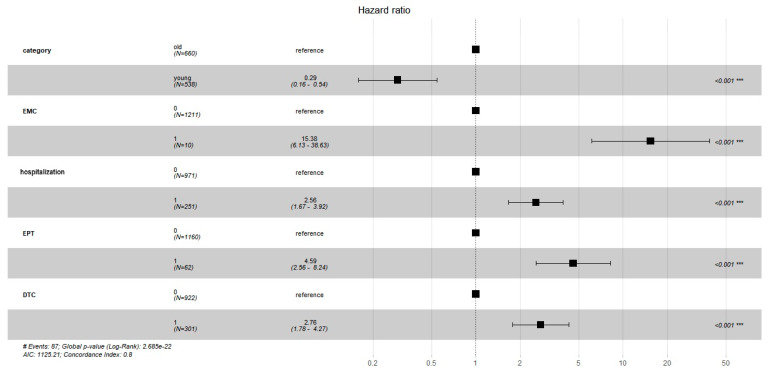
Forest plot for hazard ratio for the variables in Cox regression model. EMC: intracerebral hemorrhage (1: with; 0: without); EPT: chronic liver disease; DTC: cognitive impairment. The Cox regression model (concordance 0.8 [se 0.02]; likelihood ratio 110.9; Wald test 113; score (log-rank) 156.1; *p* < 2 × 10^−16^) shows that variables significantly correlated to 90-day mortality were age (less than 75 years old is a protective factor, HR 0.29 [95%CI 0.16–0.54]; *p* < 0.001); chronic liver disease (HR 4.59 [95%CI 2.56–8.24], *p* < 0.001); cognitive impairment (HR 2.76 [95%CI 1.78–4.27], *p* < 0.001); intracerebral hemorrhage (HR 15.38 [95%CI 6.13–38.63], *** *p* < 0.001); and hospitalization (HR 2.56 [95%CI 1.67–3.92], *p* < 0.001).

**Figure 6 geriatrics-09-00023-f006:**
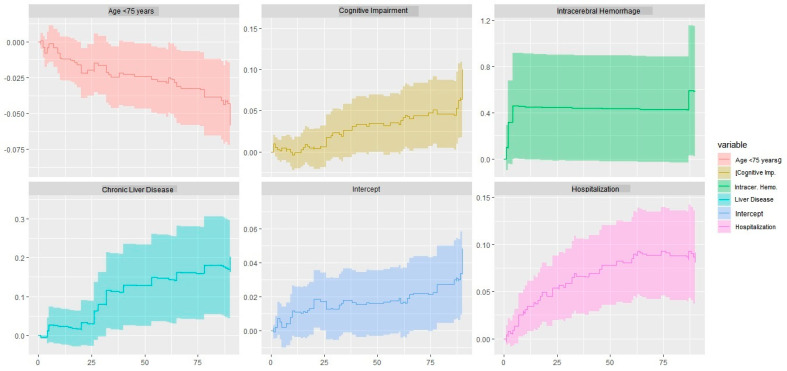
Aalen model plots. The plots show how the effects of the covariates change over time. The Aalen additive model estimates the additive contributions of differences in covariate values from reference values to the hazard of an event as functions of time. The covariate values can be fixed or time-varying. In particular, hospitalization, chronic liver disease, and cognitive impairment show an exponential increase in mortality over time. As age increases, the protective effect decreases in relation to 90-day mortality.

**Figure 7 geriatrics-09-00023-f007:**
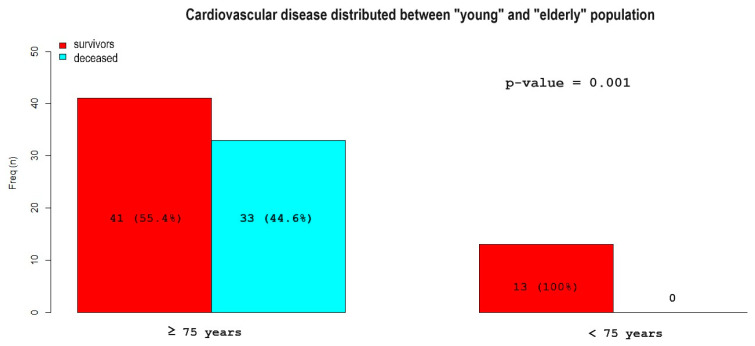
Cardiovascular disease distribution among patients < 75 years and ≥75 years. The patients who died are depicted in blue, while those who survived are depicted in red. Cardiovascular disease is more prevalent in the older population than in the younger population, both among surviving and deceased patients.

**Table 1 geriatrics-09-00023-t001:** The distribution of clinical variables with their evaluation as 90-day mortality risk factors. (INR: international normalized ratio. * pedestrian, fall from >2 m, ejected passenger).

		Alive	Deceased	Hazard Ratio (95%CI)	*p*-Value
		N = 1131	N = 90		
Demographic characteristics					
Age (years) [IQR]		77 (57–85)	88 (80–91)	1.07 (1.05–1.09)	*<0.001*
Sex (males) (%)		504 (45%)	38 (42%)	0.92 (0.60–1.40)	0.696
Clinical history					
Arterial Hypertension (%)		499 (44%)	54 (60%)	1.85 (1.21–2.82)	*0.004*
Diabetes Mellitus (%)		162 (14%)	25 (28%)	2.17 (1.37–3.44)	*0.001*
Chronic Kidney Disease (%)		111 (10%)	19 (21%)	2.38 (1.43–3.95)	*0.001*
Alcoholism (%)		83 (7%)	6 (7%)	0.90 (0.39–2.05)	0.790
Chronic obstructive pulmonary disease (%)		63 (6%)	11 (12%)	2.26 (1.20–4.25)	*0.009*
Chronic Liver Disease (%)		47 (4%)	15 (17%)	4.12 (2.37–7.18)	*<0.001*
Cardiac Disease (%)		336 (30%)	34 (38%)	1.41 (0.92–2.15)	0.116
Frequent Falls (%)		236 (21%)	38 (42%)	2.59 (1.71–3.94)	*<0.001*
Cognitive Impairment (%)		252 (22%)	49 (54%)	3.83 (2.53–5.81)	*<0.001*
Blood thinning medication					
Vitamin K antagonists (%)		112 (10%)	7 (8%)	0.77 (0.36–1.67)	0.515
Direct oral anticoagulants (%)		57 (5%)	3 (3%)	0.65 (0.21–2.06)	0.458
Low molecular weight heparin (or fondaparinux) (%)		9 (1%)	3 (3%)	3.81 (1.20–12.0)	*0.015*
Acetylsalicylic acid (%)		191 (17%)	16 (18%)	1.07 (0.62–1.83)	0.807
Clopidogrel (%)		38 (3%)	3 (3%)	1.00 (0.32–1.83)	0.998
Blood tests					
INR [IQR]		1.07 [1.00–1.26]	1.13 [1.03–1.28]	0.97 (0.60–1.56)	0.896
Platelets (count × 10^3^/mm^3^) [IQR]		218 [178–264]	226 [184–286]	1.00 (1.00–1.01)	0.131
Blood Alcohol (g/L) [IQR]		1.31 [0.10–2.84]	0.10 [0.10–0.10]	0.40 (0.12–1.39)	0.151
Level of consciousness					
Glasgow Coma Scale [IQR]		15 [15–15]	15 [15–15]	0.59 (0.28–1.21)	0.149
Symptoms					
Amnesia (%)		91 (8%)	8 (9%)	1.13 (0.55–2.34)	0.741
Loss of Consciousness (%)		92 (8%)	4 (4%)	0.54 (0.20–1.46)	0.214
Vomiting (%)		27 (2%)	1 (1%)	0.48 (0.07–3.43)	0.453
Headache (%)		85 (8%)	0 (0%)	0	*0.008*
Focal Neurological Deficit (%)		13 (1%)	1 (1%)	1.01 (0.14–7.27)	0.989
Confusion (%)		113 (10%)	23 (26%)	2.98 (1.85–4.79)	*<0.001*
High Energy Impact					
High Energy Impact * (%)		59 (5%)	1 (1%)	0.21 (0.03–1.54)	0.092
Associated injuries					
Distal Upper Limb Fracture (%)		46 (4%)	4 (4%)	1.09 (0.40–2.97)	0.866
Proximal Upper Limb Fracture (%)		43 (4%)	1 (1%)	0.29 (0.04–2.12)	0.196
Rib fractures (%)		36 (3%)	2 (2%)	0.71 (0.17–2.87)	0.626
	1 rib (%)	15 (1%)	1 (1%)		
	2 ribs (%)	10 (1%)	0 (0%)		
	≥3 ribs (%)	10 (1%)	1 (1%)		
Spine fracture (%)		25 (2%)	3 (3%)	1.55 (0.49–4.89)	0.452
Shoulder Girdle Fracture (%)		9 (1%)	1 (1%)	1.45 (0.20–10.4)	0.711
Pelvic fracture (%)		12 (1%)	3 (3%)	2.89 (0.92–9.15)	0.061
Distal Lower Limb Fracture (%)		17 (2%)	1 (1%)	0.74 (0.10–5.31)	0.765
Proximal Lower Limb Fracture (%)		35 (3%)	7 (8%)	2.52 (1.16–5.44)	*0.015*
Abdominal Trauma (%)		13 (1%)	0 (0%)	0	0.315
Facial Fracture (%)		86 (8%)	5 (6%)	0.72 (0.29–1.77)	0.471
Neurocranium Fracture (%)		14 (1%)	1 (1%)	0.92 (0.13–6.63)	0.938
Head CT scan					
Subdural Hematoma (%)		27 (2%)	3 (3%)	1.37 (0.43–4.34)	0.586
Brain Contusion (%)		17 (2%)	0 (0%)	0	0.250
Hemorrhagic Petechiae (%)		14 (1%)	1 (1%)	0.92 (0.13–6.59)	0.933
Intracerebral Hemorrhage (%)		5 (0.4%)	5 (6%)	11.5 (4.68–28.4)	*<0.001*
Subarachnoid Hemorrhage (%)		28 (2%)	1 (1%)	0.46 (0.06–3.29)	0.427
Clinical management					
Hospitalization (%)		212 (19%)	39 (43%)	3.17 (2.09–4.81)	*<0.001*
Discharge (%)		798 (71%)	51 (57%)	0.55 (0.36–0.84)	*0.004*
Abandonment (%)		88 (8%)	1 (1%)	0.14 (0.02–0.99)	*0.021*

## Data Availability

The data shall be available on request, with appropriate justification.

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
