# Peer review of "Risk Factors Analysis for 90-Day Mortality of Adult Patients with Mild Traumatic Brain Injury in an Italian Emergency Department"

_geriatrics, 2024, doi:10.3390/geriatrics9020023_

Round 1
Reviewer 1 Report
Comments and Suggestions for Authors
Dear Authors,
I have read your paper with interests - I believe You have focused on very important problem of Emergency Departments. I would be very satisfied if your valuable manuscript could be enriched with following data:
1. Could you please analyze whether any Co-morbidity "index" understood as number of co-exisitng health problems was an independent risk factor? E.g. did patients with 4 co-existing health problem had higher risk of death than patient with 2 co-existing health problem? That would allow creating some form of prognostic algorithm based upon you data
2. Could you publish data on combined therapy : anticoagulant and anti -platelet as risk factor of death after mTBI
3. Could you publish mean or median time of death after the mTBI?
4. Are you able to provide exact causes of death in mTBI patients? Ar at least percentage of causes of death the could be somehow related to trauma.
Author Response
We are grateful to Reviewer #1 for the attention he has given to our research.
1- The suggestion of Reviewer #1 is very intriguing. Unfortunately, in the terms with which the study was designed (the goal was to evaluate a population of patients with TBI, and therefore, the collection of medical records focused on the risk factors for brain injury known from the literature), we cannot calculate the Charlson Comorbidity Index.
However, we considered another variable: the number of comorbidities among those we reported in the database. We then categorized these into greater or lesser than 4 (most had 2 or 3 comorbidities). It is our understanding that this method is approximate and does not allow us to assess the various individual comorbidities.
However, we did not find any significant correlation with 90-day mortality (p-value = 0.18). We examined whether there was a correlation between age and comorbidities and the outcome of 90-day mortality. Also, in this case, we did not find a statistical significance (p-value= 0.35).
In any case, we consider this a result worthy of discussion. Therefore, we have added the following paragraph to the 'Results' section:
"There was no statistically significant correlation between mortality at 90 days and the amount of comorbidity (p-value 0.177) or indiscriminate anticoagulant or antiplatelet administration (p-value = 0.190).
We added also the following paragraph to the “Discussion” section:
“The absence of a significant correlation between the degree of comorbidity in the population is intriguing. However, this is consistent with the literature. Recently, a systematic review by Xiong et al, analyzing 27 cohort studies, found no significant correlation between the comorbidity condition load and outcome [26]. A more recent systematic review by Hanafy et al. is in agreement with this result [27]. As the authors point out, the high heterogeneity of the methods used to report the load of comorbidity conditions makes it difficult to draw conclusions. The standardization of the comorbidity degree report remains an open question in TBI studies. Unfortunately, for the study design that we have set, we are not able to better detail the degree of comorbidity of the sample we analyzed (e.g., Charlson comorbidity index, etc.). However, in our population, we observe that the comorbidities between the elderly and the young were qualitatively distinct. Alcohol-related disorders, including chronic liver disease, were more prevalent among the younger population, while cardiovascular disease was more prevalent among the older population. From a young age, alcohol consumption is linked to TBI, as evidenced by literature [28]. This is connected to a higher rate of high-energy accidents in younger patients. The elderly population, on the other hand, is more vulnerable to cardiovascular diseases and cognitive decay [29].
Otherwise, analyzing the older population, it appears that the number of comorbidities was significantly correlated with the 90-day mortality. In particular, cognitive impairment and a history of falls were factors involved in the outcome of this population. The fragility of these patients seemed to determine the outcome more than the trauma itself. According to some studies, mild TBI is a significant risk factor for death among the elderly population [30]. From a molecular viewpoint, TBI appears to be an inflammatory trigger that causes neurodegeneration [31]. It seems that the process of accelerated neuronal senescence is the cause of cognitive decline. In turn, this factor can lead to a vicious circle of falls and more traumatic events.
However, it is worth exploring the fact that age itself (even without the contribution of comorbidities) is a risk factor for mortality in the event of TBI. Ostermann et al discovered that respiratory failure was a risk factor for short-term mortality in elderly (> 65 years) TBI patients [32]. In view of the clinical fragility of this population, and in light of the role of hospitalization in determining the risk of death, it is plausible that conditions related to hospitalization (e.g., nosocomial infections) can contribute to 90-day mortality. Indeed, Selassie et al, in a retrospective study conducted on more than 41,000 patients, found that the occurrence of nosocomial infections (most of which are respiratory infections) is one of the causes of death of patients suffering from TBI [33]. However, it is a challenge to establish the epidemiology of elderly patients with mild TBI and comprehend the factors responsible for the unfortunate outcome, because the majority of current literature focuses on severe or at least moderate TBI. The population who experiences mild TBI, particularly those who are elderly, seems to be receiving little attention from clinical studies.
We found no statistically significant differences in the outcome between females and males. Although the literature seems to describe a higher incidence of TBI in elderly women, there is no evidence that this could worsen the outcome of female patients [34]. However, it can be seen that the female gender is more represented in the older population. Aging seems to cause a selection towards the female gender, without however this presents significant worse rates of trauma-specific survival [35].
We did not find any statistically significant difference in our population's intake of anticoagulants or antiplatelet therapy. The most recent systematic reviews and meta-analyses found no statistically significant difference in the mortality rate of these subgroups of patients with mild TBI [36-38]. This result is significant because many patients with mild TBI, despite the absence of any clinical signs, are subjected to CT scans and an observation period exclusively due to chronic therapy with anticoagulants [39]. However, our findings show that anticoagulant and/or antiplatelet treatment did not affect the 90-day mortality rate. A recent retrospective study in Austria found that a 1-year increase in mortality can be attributed to both anticoagulants and antiplatelets [40]. However, this study also considered severe TBI. The low mortality rate found specifically in the cohort of patients with mild TBI makes it hard to draw conclusions. Furthermore, a large study by Fahkry et al, that analyzed the geriatric population (33,710 patients), found that anticoagulation and/or antiplatelet therapy did not have a significant correlation with mortality [41]. This may indicate that, in the determinism of the outcome of this cohort of patients, other factors (such as those we have found) mostly predict an unfortunate outcome.”
2- 34% of patients (n= 416) took either an anticoagulant or an antiplatelet. At univariate regression it was found that taking at least one drug of these two categories significantly correlated with 90-day mortality (p-value = 0.19).
Our result seems to be contradictory to what is reported in the literature. The consumption of an anticoagulant or antiplatelet was linked to an increase in mortality by one year, according to a recent study in Austria. However, unlike our study, this also takes into account severe head injuries. Drawing conclusions is difficult due to the low mortality rate in the cohort of patients with mild traumatic brain injury. Expanding the sample collection would be necessary. However, the literature is discordant and, in the older population, it seems that anticoagulant or antiplatelet therapy does not play a decisive role in the outcome. We highlighted this discrepancy in the Manuscript.
We added the followin paragraph to the “Discussion” section:
“A recent retrospective study in Austria found that a 1-year increase in mortality can be attributed to both anticoagulants and antiplatelets [40]. However, this study also considered severe TBI. The low mortality rate found specifically in the cohort of patients with mild TBI makes it hard to draw conclusions. Furthermore, a large study by Fahkry et al, that analyzed the geriatric population (33,710 patients), found that anticoagulation and/or antiplatelet therapy did not have a significant correlation with mortality [41]. This may indicate that, in the determinism of the outcome of this cohort of patients, other factors (such as those we have found) mostly predict an unfortunate outcome.”
3 - We added the following sentence in the "Results" section:
“Patients with mild TBI who died within 90 days had a median survival time of 45 days.
4 - The cause of death for patients is unknown to us. Partly because patients who were not admitted were not examined, and partly because for most deceased patients, the causes were generic and not substantiated. However, 26 patients died within a week of the traumatic event. Although it is not possible to draw a causal relationship, it is plausible that the trauma is somehow involved in determining the death of these patients.
We added this limitation in “limitations” sub-section:
“The cause of death for patients cannot be determined by us. The reason is partly because the cause of death for patients who are not admitted has not been explained; and partly because even admitted patients often have generic and not sufficiently detailed causes of death. It came to our attention that 26 patients died within a week of the trauma. Despite the inability to establish a causal link, it is plausible that the trauma has played a role in determining the outcome. ”
Reviewer 2 Report
Comments and Suggestions for Authors
The paper describes novel findings linking mild traumatic brain injury and mortality. The data are complementary to other large cohort study that have evaluated mortality risk in patients with severe and mild traumatic brain injury. There are several limitations that should be addressed:
(1) It is not clear, what is the difference between the hospitalized patients and discharged patients; are they patients not hospitalized but spent some time in Emergency department? It is not clrar how many days (range) they have spent at a hospital? in addition, no discussion on "discharged" provided in thepaper although the HR values were significantly decreased; (2)Figs 5 and 6 should include only terms in English ("Ricovero" means "admission, hospitalization") and explanations for abbreviations; X/Y axis units should be explained (Fig 6); (3) it is not clear where and how the Bonferroni was employed in statistics; (4) the main conclusion most related to the scope of journal is the age as a risk factor of 90-day mortality. However, no any discussion initiated on the discovered interesting fact that less than 75 years old is a significant protective factor with HR 0.29. Whether the more than 75-old vs less than 75 years old differ in comorbidity (-ies)? Evaluation by CIRS/Charlson or at least by the most common comorbidities are required (analysis for different age groups) since, for example, it is not clear whether chronic liver disease , or cognitive impairement that associate with mortality in a whole cohort, are differently linked to mortality in 75+ vs less than 75 old patients?
The data presented in Table 1, for the item "Age (years)" which appears to be different from what is expected based on the table header. About improper Table 1 title informativity, my suggestion for the Table 1 title could be "Demographic and clinical variables as mortality risk factors" or somehow similar.
I also express my regret that the authors did not present data to compare mortality and other factors in >75 years old vs <75 years old patient groups that could improve the quality of the paper to better fit the scope of the journal.
Author Response
We are grateful to Reviewer #2 for the attention he has given to our research.
1 - Patients in our sample were either hospitalized or discharged after an observation period of usually 6-12 hours. Hospitalization is reserved for patients who have been admitted to the ward. This definition has been specified in the section titled 'Study variables':
“The patient's admission to a hospital ward or discharge after an observation period in the emergency department (generally 6 - 12 hours) was also noted. “Hospitalization” is reserved for patients who have been admitted to the ward.”
2 – We have fix it.
3 - Bonferroni multiplicity correction has been used in Cox’s multivariable model. We have specified that in Methods.
4 – It is a very interesting point that Reviewer #2 raises. With regard to the contribution of comorbidities, we did not find a correlation with the considered outcome. And yet, the link between age and mortality remains worth exploring.
We have therefore made a comparison between the two sub-populations (greater or less than 75 years). The conclusions we reached are reported in a dedicated sub-section of "Results":
“Age-based population analysis
Analyzing the two sub-populations established through cut-off by age corresponding to 75 years, we found that the elderly population has a greater number of female individuals (66% vs. 43%; p <0.001), a greater rate of comorbidity (hypertension: 66% vs 21%, p < 0.001; diabetes mellitus: 21% vs 9%, p <0.001; chronic kidney disease: 18% vs 4%; p < 0.001; chronic obstruction pulmonary disease: 8% vs 4%; p = 0.005; cardiovascular disease: 47% vs 11%; p <0.001; cognitive impairment 36% vs 10%; p <0.001; and more history of falls: 30% vs 13%; p <0.001); a higher proportion of patients taking anticoagulants and/or antiplatelet drugs (50% vs. 16%; p <0.001), a higher INR median value (1.12 [95%CI: 1.03 – 1.56] vs. 1.03 [95%CI 0.97 – 1.09]; p <0.001), more frequently proximal fractures of the limbs (5% vs 2%; p = 0.026 for the superior limbs; 5% vs 2%; p = 0.002 for the inferior limbs), and confusion (14% vs 7%, p <0.001), than the younger population. On the other hand, the younger population is more likely to have a history of chronic alcoholism (15% vs 1%; p <0.001). They also more often experience headache (13% vs 2%, p <0.001) and loss of consciousness (11% vs 5%, p <0.001). Finally, more often a high-energy trauma (9% vs 2%, p <0.001).
Analyzing the older population, it is noted that patients who died at 90-days, are significantly older (87 [95%CI 82-91] vs 84 [95%CI 81-89]; p <0.001; RR 1.04); they have a greater number of comorbidities (3 [95%CI 2-4] vs 2 [95%CI 1-3]; p <0.001), in particular a greater rate of cognitive deterioration (55% vs 34%; p <0.001; RR 2.17), chronic liver disease (12% vs 3%; p = 0.002; RR 3.24), and history of falls (43% vs 28%; p = 0.011; RR 1.79). In addition, they have more often confusion (22% vs 12%; p = 0.002; RR 1.72) during presentation.
However, there are some differences in variables when comparing only those who have died to the two sub-populations of young and old (74 vs 13 deaths, respectively). The incidence of chronic liver disease (39% vs 12%; p = 0.032; RR 3.26) and chronic alcoholism (39% vs 0%; p <0.001; RR 10.2) is more prevalent in the younger population. The elderly are more susceptible to hypertension (70% vs 15%; p <0.001; RR 0.11) and cardiovascular disease (45% vs 0%; p = 0.001; RR 0) (Figure 7).”
We have expanded the 'Discussion' section by highlighting how the elderly population subjected to mild TBI is not often considered in clinical studies.:
“The absence of a significant correlation between the degree of comorbidity in the population is intriguing. However, this is consistent with the literature. Recently, a systematic review by Xiong et al, analyzing 27 cohort studies, found no significant correlation between the comorbidity condition load and outcome [26]. A more recent systematic review by Hanafy et al. is in agreement with this result [27]. As the authors point out, the high heterogeneity of the methods used to report the load of comorbidity conditions makes it difficult to draw conclusions. The standardization of the comorbidity degree report remains an open question in TBI studies. Unfortunately, for the study design that we have set, we are not able to better detail the degree of comorbidity of the sample we analyzed (e.g., Charlson comorbidity index, etc.). However, in our population, we observe that the comorbidities between the elderly and the young were qualitatively distinct. Alcohol-related disorders, including chronic liver disease, were more prevalent among the younger population, while cardiovascular disease was more prevalent among the older population. From a young age, alcohol consumption is linked to TBI, as evidenced by literature [28]. This is connected to a higher rate of high-energy accidents in younger patients. The elderly population, on the other hand, is more vulnerable to cardiovascular diseases and cognitive decay [29].
Otherwise, analyzing the older population, it appears that the number of comorbidities was significantly correlated with the 90-day mortality. In particular, cognitive impairment and a history of falls were factors involved in the outcome of this population. The fragility of these patients seemed to determine the outcome more than the trauma itself. According to some studies, mild TBI is a significant risk factor for death among the elderly population [30]. From a molecular viewpoint, TBI appears to be an inflammatory trigger that causes neurodegeneration [31]. It seems that the process of accelerated neuronal senescence is the cause of cognitive decline. In turn, this factor can lead to a vicious circle of falls and more traumatic events.
However, it is worth exploring the fact that age itself (even without the contribution of comorbidities) is a risk factor for mortality in the event of TBI. Ostermann et al discovered that respiratory failure was a risk factor for short-term mortality in elderly (> 65 years) TBI patients [32]. In view of the clinical fragility of this population, and in light of the role of hospitalization in determining the risk of death, it is plausible that conditions related to hospitalization (e.g., nosocomial infections) can contribute to 90-day mortality. Indeed, Selassie et al, in a retrospective study conducted on more than 41,000 patients, found that the occurrence of nosocomial infections (most of which are respiratory infections) is one of the causes of death of patients suffering from TBI [33]. However, it is a challenge to establish the epidemiology of elderly patients with mild TBI and comprehend the factors responsible for the unfortunate outcome, because the majority of current literature focuses on severe or at least moderate TBI. The population who experiences mild TBI, particularly those who are elderly, seems to be receiving little attention from clinical studies.
We found no statistically significant differences in the outcome between females and males. Although the literature seems to describe a higher incidence of TBI in elderly women, there is no evidence that this could worsen the outcome of female patients [34]. However, it can be seen that the female gender is more represented in the older population. Aging seems to cause a selection towards the female gender, without however this presents significant worse rates of trauma-specific survival [35].
We did not find any statistically significant difference in our population's intake of anticoagulants or antiplatelet therapy. The most recent systematic reviews and meta-analyses found no statistically significant difference in the mortality rate of these subgroups of patients with mild TBI [36-38]. This result is significant because many patients with mild TBI, despite the absence of any clinical signs, are subjected to CT scans and an observation period exclusively due to chronic therapy with anticoagulants [39]. However, our findings show that anticoagulant and/or antiplatelet treatment did not affect the 90-day mortality rate. A recent retrospective study in Austria found that a 1-year increase in mortality can be attributed to both anticoagulants and antiplatelets [40]. However, this study also considered severe TBI. The low mortality rate found specifically in the cohort of patients with mild TBI makes it hard to draw conclusions. Furthermore, a large study by Fahkry et al, that analyzed the geriatric population (33,710 patients), found that anticoagulation and/or antiplatelet therapy did not have a significant correlation with mortality [41]. This may indicate that, in the determinism of the outcome of this cohort of patients, other factors (such as those we have found) mostly predict an unfortunate outcome.”
Furthermore, we have change the title of Table 1, according to Reviwer#2's suggestion.
Reviewer 3 Report
Comments and Suggestions for Authors
This is an interesting retrospective cohort study of 1221 patients, median age 78 years, range 58-85 years, with a 90-day mortality rate of 7 % (90 patients). A multivariable Cox proportional hazards survival analysis was conducted to detect variables significantly related to 90-day mortality: age, cognitive impairment, intracerebral hemorrhage, hospitalization and chronic liver disease. In univariate survival analysis more variables were found: age, hypertension, diabetes mellitus, chronic kidney disease, chronic obstructive pulmonary disease, liver disease, frequent falls, confusion, and cognitive impairment. Intake of anticoagulants or antiplatelet drugs was not a statistically different variable.
I have a few remarks:
1) age distribution. The selected patients were adults, > 18 years. It is remarkable that in this consecutive series the age range was 58-85 years. There were no patients of 19-57 years old. It should be more discussed.
2) 90 patients died, median age 88 years (compared to surviving patients: 77 years). These patients often had multiple diseases, like arterial hypertension (n-54), cognitive impairment (n=49), frequent falls (n=38), and cardiac disease (n=34). In the total population of 1221 patients, 30 cases had a subdural hematoma. However only 3 of the deceased patients had a subdural hematoma. This number, and the fragile population with multiple diseases, may explain why no patients with documented brain lesions underwent neurosurgery. This, should be more explained, discussed in the text.
3) One-hundred twenty-one patients were positive at the head CT-scan (page 3, line 108-109). In table 1 15 had a neurocranium fracture, 30 a subdural hematoma, 17 a brain contusion, 15 hemorrhagic petechiae, 10 an intracerebral hemorrhage and 29 a subarachnoid hemorrhage. That makes 116, not 121.
Minor comments:
- The legends of Fig 5 and 6: the term ricovero is used, which is an Italian term
- Page 11 of 13, line 169: highest 90-mortality should be highest 90-day mortality
- Splanchnocranium fracture (page 6, table 1) is not a well-known term for the readers of Geriatrics, facial fracture?
Author Response
We are grateful to Reviewer #3 for the attention he has given to our research.
1 - The reviewer's note is intriguing. We added the following sentence to the "limitations" sub-section:
“Although adult patients aged 18 and over were included in the data collection, there were no representatives from 18-57. Younger people are likely to be more susceptible to moderate or severe TBI than older individuals.”
2 - We expanded the 'Discussion' section based on the reviewers' suggestions, discussing the role that the fragility of the elderly population and the burden of comorbidity play in mortality. In addition, specifically regarding the low rate of surgery for cases of subdural hematoma, we added the following phrases:
“Patients with a documented brain lesion are hospitalized more frequently than patients with a negative head CT scan [10]. The rate of hospitalization in our population was 20%. Although direct damage from head trauma may have been the cause of hospitalization, none of the hospitalized patients underwent neurosurgery. While there were 30 cases of subdural hematoma found, only 3 of them died within 90 days, even though none of them had undergone neurosurgery. Small case studies in the literature show that patients with subdural hematoma from ground-level falling have a higher mortality rate than those with high-energy trauma cases [15]. The connection between non-operational issues caused by the fragility of the elderly population or ageism is uncertain. The selection of patients for surgery is a controversial aspect that has a significant clinical impact. ”
3 - The Reviewer is correct. There was a mistake in the text we made. We corrected (116 cases).
Minor comments:
We corrected the terms in figure 5 and 6. We corrected all mistakes highlighted by the Reviewer.
Reviewer 4 Report
Comments and Suggestions for Authors
Manuscript “Risk factors analysis for 90-day mortality of adult patients 2 with mild traumatic brain injury in an Italian emergency” by Daniele Orso et al. did a retrospective cohort study on 1221 mild adult traumatic brain injury (TBI) patients collected from the Emergency Department of the University Hospital of Trieste, Italy with the aim to find out the most prominent risk factors for mortality 90 days after the injury. They found that age, history of chronic liver disease, cognitive impairment along with intracerebral haemorrhage and hospitalization were the risk factors related to 90-day mortality. These results may give some useful information for related clinicians when treating patients with mild TBI. In general, the research was well designed, and the results were objective. However, the Introduction and Discussion need to be expanded, and the presentation of the results needs to be improved. I have following comments for improvement of the presentation.
1. The Introduction needs to expand in certain extend. For example, what the worldwide prevalence of TBI is, not just limited to US; the definition of severe TBI, and mild TBI, etc.
2. Delete “Study setting and population” on line 58. It is a repetition.
3. In section 2.3, Study Variables, please write the complete name of COPD, VKA, DOACs, INR, etc. when they were used first time. Although they may probably be commonly used abbreviations in clinic but not for the readers who are not familiar with these words. Some of them were explained in Table 1, but that is not enough.
4. In Figure 1 legend, line 105, please give a complete name of IQR. I assume it indicates “Interquartile Range”.
5. In Figure 1 legend, line 108-109, please explain what “positive at head CT scan” include.
6. Table 1 needs to be improved.
First, put the title in a separate line, not together with those detailed names (COPD: chronic obstructive pulmonary disease; ……). Put these names in a note above or below the table.
Second, design the table a bit better. For example, in the first raw only p-value represents the value of corresponding column. Otherwise, they are not the case.
|
Population |
Alive |
Deceased |
Hazard Ratio (95%CI) |
p-value |
In most cases, the numbers in the parentheses were not defined. Please indicate what the numbers in the parentheses mean.
7. In general, the quality of figure 2-6 are poor. The space in the figure can be better used, and the text can be a bit larger so that they can be easily read.
8. In Figure 5, what do the different columns mean?
9. The Discussion needs a bit thorough and deeper. Discuss why different studies have different results, especially to compare with your results. For example, when citing results from other studies you cannot just mention “Our finding is in line with the recent literature [6-8]”. You need to mention what are the results from other studies, what are yours, and discuss the reasons that might cause the differences. Similar case is like “Popp et al. found greater mortality”. How greater was from their study?
10. Line 188, what does “a worsening neurological examination” mean?
11. For supplemental figures (1-3), if you want to publish them, you need to give them figure legends.
Author Response
We are grateful to Reviewer #4 for the attention he has given to our research.
1- The suggestion is very useful, and we have extended the Introduction with the following part:
“Traumatic brain injury (TBI) is characterized by an external force that causes an alteration in brain function or other evidence of brain pathology [1]. TBI is estimated to cause an estimated 10 million deaths or hospitalizations worldwide each year [2]. More than 1.7 million cases of TBI occur annually in the United States. About 75% of this group is classified as mild TBI [3].
TBI has traditionally been classified using the Glasgow Coma Scale (GCS), a clinical score that measures consciousness level. In general, mild TBI is diagnosed in patients with a GCS score of 13–15; moderate TBI is diagnosed in those with a GCS score of 9–12; and severe TBI is diagnosed in those with a GCS score <8 [2].”
2 – We have fix it.
3 – We have corrected it.
4 – We have corrected it.
5 - We specified what we meant by positive CT scan.
6 – 8 - We have modified Table 1 as per the Reviewer's instructions. Regrettably, we have not been able to improve the figures. We added a Figure 7 to better explain the sub-analysis based on the age of the patients.
9 - We have rewritten much of the Discussion. We have more detailed references to the present literature. We also expanded the discussion on comorbidities and age as follows:
“In our cohort of patients with mild TBI, 90-day mortality was 7%. Our finding is in line with the recent literature [8-10]. Grosswasser et al., in a small cohort (279 patients) of TBI victims followed for a 5-year period, reported a mortality rate of 11.5% [9]. Leitner et al. reported a 30-day mortality rate of 2.5% in a population similar to ours (1,788 patients), but almost 6% in the population over the age of 65 [10] […]
The absence of a significant correlation between the degree of comorbidity in the population is intriguing. However, this is consistent with the literature. Recently, a systematic review by Xiong et al, analyzing 27 cohort studies, found no significant correlation between the comorbidity condition load and outcome [26]. A more recent systematic review by Hanafy et al. is in agreement with this result [27]. As the authors point out, the high heterogeneity of the methods used to report the load of comorbidity conditions makes it difficult to draw conclusions. The standardization of the comorbidity degree report remains an open question in TBI studies. Unfortunately, for the study design that we have set, we are not able to better detail the degree of comorbidity of the sample we analyzed (e.g., Charlson comorbidity index, etc.). However, in our population, we observe that the comorbidities between the elderly and the young were qualitatively distinct. Alcohol-related disorders, including chronic liver disease, were more prevalent among the younger population, while cardiovascular disease was more prevalent among the older population. From a young age, alcohol consumption is linked to TBI, as evidenced by literature [28]. This is connected to a higher rate of high-energy accidents in younger patients. The elderly population, on the other hand, is more vulnerable to cardiovascular diseases and cognitive decay [29].
Otherwise, analyzing the older population, it appears that the number of comorbidities was significantly correlated with the 90-day mortality. In particular, cognitive impairment and a history of falls were factors involved in the outcome of this population. The fragility of these patients seemed to determine the outcome more than the trauma itself. According to some studies, mild TBI is a significant risk factor for death among the elderly population [30]. From a molecular viewpoint, TBI appears to be an inflammatory trigger that causes neurodegeneration [31]. It seems that the process of accelerated neuronal senescence is the cause of cognitive decline. In turn, this factor can lead to a vicious circle of falls and more traumatic events.
However, it is worth exploring the fact that age itself (even without the contribution of comorbidities) is a risk factor for mortality in the event of TBI. Ostermann et al discovered that respiratory failure was a risk factor for short-term mortality in elderly (> 65 years) TBI patients [32]. In view of the clinical fragility of this population, and in light of the role of hospitalization in determining the risk of death, it is plausible that conditions related to hospitalization (e.g., nosocomial infections) can contribute to 90-day mortality. Indeed, Selassie et al, in a retrospective study conducted on more than 41,000 patients, found that the occurrence of nosocomial infections (most of which are respiratory infections) is one of the causes of death of patients suffering from TBI [33]. However, it is a challenge to establish the epidemiology of elderly patients with mild TBI and comprehend the factors responsible for the unfortunate outcome, because the majority of current literature focuses on severe or at least moderate TBI. The population who experiences mild TBI, particularly those who are elderly, seems to be receiving little attention from clinical studies.
We found no statistically significant differences in the outcome between females and males. Although the literature seems to describe a higher incidence of TBI in elderly women, there is no evidence that this could worsen the outcome of female patients [34]. However, it can be seen that the female gender is more represented in the older population. Aging seems to cause a selection towards the female gender, without however this presents significant worse rates of trauma-specific survival [35].
We did not find any statistically significant difference in our population's intake of anticoagulants or antiplatelet therapy. The most recent systematic reviews and meta-analyses found no statistically significant difference in the mortality rate of these subgroups of patients with mild TBI [36-38]. This result is significant because many patients with mild TBI, despite the absence of any clinical signs, are subjected to CT scans and an observation period exclusively due to chronic therapy with anticoagulants [39]. However, our findings show that anticoagulant and/or antiplatelet treatment did not affect the 90-day mortality rate. A recent retrospective study in Austria found that a 1-year increase in mortality can be attributed to both anticoagulants and antiplatelets [40]. However, this study also considered severe TBI. The low mortality rate found specifically in the cohort of patients with mild TBI makes it hard to draw conclusions. Furthermore, a large study by Fahkry et al, that analyzed the geriatric population (33,710 patients), found that anticoagulation and/or antiplatelet therapy did not have a significant correlation with mortality [41]. This may indicate that, in the determinism of the outcome of this cohort of patients, other factors (such as those we have found) mostly predict an unfortunate outcome.”
10 - We corrected it as follows:
“In the literature, a worsening neurological status appear to be the major predictor for a second head CT scan with pathological findings”
11- The additional figures were removed because we thought they were neither explanatory nor essential.
Round 2
Reviewer 4 Report
Comments and Suggestions for Authors
All my comments were satisfactorily addressed. I don't have any further comments.
Author Response
We thank the Reviewers for their attention to us. The paper has improved considerably after the suggestions made.